# Electronic Interactions Between the Receptor-Binding Domain of Omicron Variants and Angiotensin-Converting Enzyme 2: A Novel Amino Acid–Amino Acid Bond Pair Concept

**DOI:** 10.3390/molecules30092061

**Published:** 2025-05-06

**Authors:** Puja Adhikari, Bahaa Jawad, Wai-Yim Ching

**Affiliations:** 1Department of Physics and Astronomy, University of Missouri-Kansas City, Kansas City, MO 64110, USA; bahaa.a.jawad@uotechnology.edu.iq (B.J.); chingw@umkc.edu (W.-Y.C.); 2College of Applied Sciences, University of Technology, Baghdad 10066, Iraq

**Keywords:** SARS-CoV-2, omicron variant (BA.1, BA.2, BA.5, and XBB.1.16), density functional theory, amino acid–amino acid bond pair

## Abstract

SARS-CoV-2 remains a severe threat to worldwide public health, particularly as the virus continues to evolve and diversify into variants of concern (VOCs). Among these VOCs, Omicron variants exhibit unique phenotypic traits, such as immune evasion, transmissibility, and severity, due to numerous spike protein mutations and the rapid subvariant evolution. These Omicron subvariants have more than 15 mutations in the receptor-binding domain (RBD), a region of the SARS-CoV-2 spike protein that is important for recognition and binding with the angiotensin-converting enzyme 2 (ACE2) human receptor. To address the impact of these high numbers of Omicron mutations on the binding process, we have developed a novel method to precisely quantify amino acid interactions via the amino acid–amino acid bond pair (AABP). We applied this concept to investigate the interface interactions of the RBD–ACE2 complex in four Omicron Variants (BA.1, BA.2, BA.5, and XBB.1.16) with its Wild Type counterpart. Based on the AABP analysis, we have identified all the sites that are affected by mutation and have provided evidence that unmutated sites are also impacted by mutation. We have calculated that the binding between RBD and ACE2 is strongest in OV BA.1, followed by OV BA.2, WT, OV BA.5, and OV XBB.1.16. We also present the partial charge values for all 311 residues across these five models. Our analysis provides a detailed understanding of changes caused by mutation in each Omicron interface complex.

## 1. Introduction

Interaction among biomolecules is a complicated issue of inherent biological system interest [1]. A thorough and quantitative knowledge of these interactions at the atomic level, utilizing more advanced computational methods, is essential not just from a fundamental viewpoint, but also to guide the design of better therapy. Amino acids (AAs) are the building blocks of a protein and hence are the focus point in studying biomolecular interactions. An assembly of alpha carbon atoms, an amino group, a carboxyl group, hydrogen atoms, and side chains form the AA. A protein is composed of a long chain of these AAs, each covalently connected to its neighbor by a peptide bond. Proteins have four hierarchical structures: primary, secondary, tertiary, and quaternary structures. The linear sequence of AAs within a protein is considered as the primary structure. The local folding of long-chain AAs in proteins, driven by hydrogen bonds between backbone atoms, forms secondary structures such as the α-helix, β-sheet, and random coils. A tertiary structure is generated when this secondary structure is folded further due to disulfide bridges, hydrogen bonds, and hydrophobic, hydrophilic, and electrostatic interactions. The quaternary structure is the result of the spatial arrangement of different tertiary subunits. The protein folds into unique three-dimensional (3D) networks that serve critical roles in its function. Thousands of bonds—covalent and non-covalent—and hydrogen bonding hold the protein together.

Protein–protein interactions (PPIs) are very important for understanding cellular functions, disease mechanisms, and drug design [2]. While experimental techniques provide valuable insights into PPIs, they are time-consuming and labor intensive. To overcome these limitations, several computational methods have been developed to predict PPIs efficiently. These methods include protein sequence-based methods, comparative genomics-based methods [2,3], gene co-expression-based methods [2,4], function-based methods [2,5], network-based methods [2,6], and structure-based methods [2,7]. Among these methods, structure-based methods have the ability to provide detailed and accurate interaction predictions at the atomic level.

Structure-based methods analyze 3D protein structures and docking models to determine potential interactions [2]. By using structural data, these methods can identify binding sites, predict interactions mechanisms, and offer insights that are particularly valuable for drug discovery and protein engineering [2]. Unlike sequence- or network-based approaches, which rely on indirect correlation patterns, structure-based methods offer a direct representation of molecular interactions. Their high accuracy makes them useful in studying PPIs. Despite their advantages, this method comes with challenges. They require high-resolution structural data, which may not always be available. In addition, they are computationally expensive. To overcome these limitations, there are machine learning-based studies trying to overcome its expense [8]. Our approach falls under structure-based method, utilizing ab initio quantum mechanical calculations to study such protein interactions. This study specially focuses on inter-amino acid interactions, which encode critical structural information about the twists and turns of the protein—a novel concept in PPI analysis. Further details on our method and the inter-amino acid bonding, termed as an amino acid–amino acid bond pair (AABP), are discussed in the Methods section.

Since the onset of the severe acute respiratory syndrome coronavirus 2 (SARS-CoV-2) pandemic, the world has faced unprecedented challenges. SARS-CoV-2 is a single positive-strand RNA virus composed of four structural proteins: spike (S), envelope (E), membrane (M), and nucleocapsid (N). Among these, the S-protein plays a pivotal role in the virus’s infectivity and transmission by initiating the initial interaction with the human cell’s angiotensin-converting enzyme-2 (ACE2) [9]. The S-protein is made up of two main parts: the S1 subunits, which help the virus attach to the host cell, and the S2 subunit, which helps the virus fuse with the cell membrane. The S1 subunit includes an N-terminal domain, receptor-binding domain, subdomain 1, and subdomain 2. The S2 subunit contains fusion peptide, heptad repeat 1, central helix, connector domain, heptad repeat 2, transmembrane domain, and a cytoplasmic tail. Among these domains, the receptor-binding domain (RBD) interacts directly with human ACE2, facilitating the infection process (see Figure 1).

Angiotensin-converting enzyme 2 (ACE2) plays an important role in the renin–angiotensin–aldosterone system (RAAS), where it helps to balance the effects of another enzyme called ACE [10]. ACE converts angiotensin I (Ang I) into a molecule called angiotensin II (Ang II), which causes blood vessels to narrow and promote inflammation and tissue damage. In contrast, ACE2 breaks down Ang II into angiotensin-(1–7) [Ang (1–7)]. Ang (1–7) has protective effects, such as relaxing blood vessels, reducing inflammation, and preventing tissue damage [10]. ACE2 can also convert Ang I into angiotensin-(1–9) [Ang (1–9)], which may lower blood pressure and protect the heart, either on its own or after being further turned into Ang (1–7). These actions of ACE2—especially through the ACE2/Ang (1–7)/Mas receptor pathway—help protect the heart, blood vessels, and kidneys [10].

However, ACE2 is also known to be an entry point for the SARS-CoV-2 virus. When the S-protein of the virus binds to ACE2, it reduces the amount of ACE2 available on the surface of cells. This leads to an imbalance in the RAAS, with higher levels of Ang II and lower levels of Ang (1–7), which may worsen inflammation and organ damage during COVID-19 [10].

The emergence of various variants of concerns (VOCs) is a new era in the evolution of the virus, which poses a serious global threat to individual health. To date, the World Health Organization (WHO) has designated five VOCs: Alpha, Beta, Gamma, Delta, and Omicron [11]. These variants exhibit distinct traits from the original Wild Type (WT) strain, affecting infectivity, antigenicity, and transmissibility [12]. As a result, infections and fatality rates have increased globally. Among these VOCs, the Omicron variant (OV) introduces multiple mutations throughout the S-protein, particularly in the RBD. The arrival of the OV BA.1 significantly altered the course of the pandemic, as it diverged substantially from the earlier variants [13], resulting in lower pathogenicity [14,15,16,17,18], higher transmissibility [15], and stronger immune evasion [19,20,21,22,23,24,25,26]. The Omicron lineage has continued to evolve, producing subvariants with even greater immune escape.

Along with having a large number of mutations, the OV has many sub-lineages, including BA.2, BA.3, BA.4/BA.5, XBB, etc. [27]. Specifically, after BA.2 was discovered in European and Asian countries in February 2022, it quickly become a predominant variant, with a growth advantage over BA.1 in terms of transmissibility, immune evasion, and infectivity [28]. By June 2022, BA.5 became a dominant variant in most countries [29]. BA.5, a sub-lineage of the OV (Pango lineage B.1.1.529), was first identified in South Africa through genomic surveillance. It differs from earlier Omicron sub-lineages (BA.1 and BA.2) due to specific mutations in the receptor-binding domain (RBD), which plays a crucial role in immune escape and infectivity [30,31,32,33]. XBB.1.16, a subvariant of the OV, emerged in early 2023 after recombining two BA.2-derived subvariants, BJ.1 and BM.1.1.1 [13,34,35]. It quickly spread through India and other countries [36,37,38]. This subvariant is notable for its enhanced immune escape, making it more capable of infecting individuals even after vaccination [39,40,41].

The interaction between spike protein RBD and the ACE2 receptor is the initial step in the viral infection, playing a crucial role in infecting tissues within a host [42]. This interaction triggers the fusion of the viral and cell membranes [43]. The binding strength between the RBD and ACE2, along with the cleavage process, significantly impacts the infectivity and transmissibility of SARS-CoV-2 [42,44]. Given this essential role of the RBD–ACE2 complex, it has been a key target for developing therapies such as vaccines, antibodies, and small-molecule inhibitors [12,45]. As a result, intensive efforts to target this interaction led to the rapid development of various vaccines and antibodies. Hence, studying the RBD–ACE2 interface across different VOCs is crucial, since it changes the interaction between them, transpiring into higher infectivity and transmissibility.

In this study, we present an analysis of the RBD–ACE2 interface across various OV subvariants, including BA.1, BA.2, BA.5, and XBB.1.16. We have thoroughly investigated amino acid–amino acid interactions for the RBD–ACE2 models, providing insights into the changes due to mutation. In the following sections, we first discuss the results, followed by the modeling of five interface models, methods utilized, and concluding remarks.

## 2. Results and Discussion

### 2.1. Amino Acid–Amino Acid Bond Pair (AABP) for Mutated Sites

We calculated the AABP values for all AAs in the RBD–ACE2 interface across five models. To simplify the analysis, we begin by focusing on the mutation sites listed in Table 1. In the mutated models for OV BA.1, OV BA.2, OV BA.5, and OV XBB.1.16, some mutations are shared across variants, while others are specific to individual variants. All such mutations have been included in the analysis. Figure 2 illustrates the AABP analysis for all mutation sites. It presents the AABP analysis for nearest neighbor (NN) interactions (Figure 2a), non-local (NL) amino acid interactions (Figure 2b), and hydrogen bonding, including O⋯H (Figure 2c) and N⋯H (Figure 2d) interactions. In Figure 2a, we see some changes in all the sites. To provide clarity in the results, we have presented the population standard deviation for all outcomes of Figure 2 in Figure 3.

From standard deviation for nearest neighbor, we can see significant changes in sites 375 and 376 (see Figure 3a), and this is due to higher NN contributions in WT and OV BA.1 compared to other OV mutations (see Figure 2a). Even though all other OV mutations have the same mutation S375F (see Table 1), OV BA.1 does not have mutation T376A (where 376 is NN of 375). This absence of T376A contributes to OV BA.1 having higher NN AABP value, like the WT. One reason may be that the hydroxyl group on threonine is missing when it is replaced by alanine at 376, making less contact with the neighbors. This substitution may promote an α-helix structure over a β-sheet structure at this position, as observed in other proteins [46,47]. This mutation may have further implications [12]. Some other sites with relatively higher standard deviation in NN AABP are 445, 460, and 493 (see Figure 3a). In site 445, both WT and OV XBB.1.16 have higher NN AABP compared to other OV variants. One interesting fact is that only XBB.1.16 has the V445P mutation. In site 460, OV XBB.1.16 has slightly lower NN AABP, as this is the one with mutation N460K, which is not common with other OV variants. In site 493, WT and BA.5 have lower NN AABP compared to others, whereas BA.1 and BA.2 have the Q493R mutation.

Although NN interactions significantly contribute to AABPs, NL interactions remain essential and cannot be overlooked. AABP values of NL AAs are shown in Figure 2b, with its standard deviation presented in Figure 3b.

Mutations at 417 and 440 have significantly higher standard deviations, as shown in Figure 3b. At 417, the WT K417 has significantly higher NL AABP, whereas N417 in all OV variants have lower NL AABP. This observation could explain why the K417N mutation leads to decreased RBD–ACE2 binding [48]. At 440, WT N440 and OV XBB.1.16 K440 have lower NL AABP, whereas N440 of OV BA.1, BA.2, and BA.5 have higher NL AABP, as shown in Figure 2b. Both WT N440 and OV XBB 1.16 K440 only interact with two NL AAs, whereas the other OV interacts with three to four NL AAs (see Appendix A). Additionally, the AABPU representations for site 440 across all five models are shown in Appendix A. These mutations at 417 and 440 are perfect examples of how the same mutation can have different results. In sequence number 417, all OVs have similar NL AABP, whereas in 440 there is a dissimilarity in OV XBB.1.16.

In all OV interface complexes, S477N mutations increase NL AABP, while E484A, N501Y and Y505H mutations decrease it. Mutation N460K is only observed in OV XBB.1.16, which significantly increases the NL AABP. As we will see in the next paragraph, hydrogen bond contributions play an important role in determining how these mutations affect interface interactions.

Now, looking into hydrogen bonding O⋯H, some of the noticeable standard deviations are in sites 376, 405, 408, 477, 484, 493 (see Figure 3c). Site 376 has the highest standard deviation since the OV BA.2, BA.5, and XBB.1.16 have mutation A376, which loses the hydroxyl group on T376 of WT and OV BA.1, leading to lower O⋯H contribution. Sites 405 and 408 also have similar cases as in site 376, with mutations only in OV BA.2, BA.5, and XBB.1.16. In site 477, all OVs have same mutation, N477, which leads to an increase in interacting NL AAs (see Appendix A) resulting in higher O⋯H AABP. Site 484 is also like site 477; however, in all OVs, A484 reduces interacting NL AAs to two from four (see Appendix A), resulting in a decrease in O⋯H AABP.

The mutation R493 results in higher O⋯H AABP values in OV BA.1 and OV BA.2 interface complexes. However, even in the absence of this mutation, Q493 residue in OV BA.5 and XBB.1.16 exhibits lower AABP values relative to WT (see Figure 2c), contributing to overall higher standard deviation (see Figure 3c). Both BA5 and XBB.1.16 have lower NL AABP in Q493 compared to WT, indicating that the reduction in O⋯H in BA5 and XBB.1.16 stems NL bonding. In XBB.1.16, Q493 interacts with S490 and also forms an interaction with K31 of ACE2, both of which are absent in WT. These unique interactions to XBB.1.16 may contribute to its reduced NL AABP from O⋯H. Additionally, Q493 in XBB.1.16 exhibits weaker interactions with E35 of ACE2 compared to WT (see Figure 4a). Figure 4 shows AABP interaction between RBD and ACE2 and will be discussed in Section 2.3.

For BA.5, Q493 lacks interaction with E35 of ACE2, whereas WT maintains a strong interaction with E35 (see Figure 4a). Furthermore, WT Q493 has stronger interaction with Y453, while the interaction of Y453 with Q493 of BA.5 is comparatively weaker. These factors collectively explain the overall lower NL and O⋯H AABP values of Q493 in BA.5 and XBB.1.16. The numbers of NL AAs are listed in Appendix A. The mutation R493 interacts with 11 NL AAs, while Q493 residue in WT, OV BA.5, and XBB.1.16 interacts with eight, seven, and nine NL AAs, respectively. The Q493R mutation is expected to enhance interface contact, which is consistent with previous studies [49,50].

A higher number of NL AAs is generally associated with higher AABP values. However, this relationship can vary, as a smaller number of NL AAs with stronger interaction may still result in higher AABP values. Another important factor is that the same mutation does not always result in the same number of NL AA interactions. This is due to several factors, including the position, orientation, and conformation of the mutation on the RBD, which may contribute to the specific NL AABP. Some NL AAs may have mutated in certain cases but not in others. These dynamics between AABP are complex yet fascinating.

Moving forward to N⋯H, site 505 has the largest standard deviation (see Figure 3d). This site has the Y505H mutation across all OVs, but they vary in N⋯H AABP values (see Figure 2d). It interacts with nine NL AAs in OV BA.1 and BA.2, and ten in OV BA.5 and OV XBB.1.16 (see Appendix A). By adding the NN and NL AABP, we would obtain the total AABP (TAABP). From Appendix A, the TAABP values in site 505 for WT, OV BA.1, OV BA.2, OV BA.5, and OV XBB.1.16. are 1.34 e^−^, 1.11 e^−^, 1.14 e^−^, 1.18 e^−^, and 1.12 e^−^, respectively. The distinction between unmutated and mutated sites is evident, as WT exhibits a higher TAABP compared to the mutated site (see Appendix A). According to this observation, the Y505H mutation causes decreased interface interaction with ACE2, which is in line with earlier research [50].

### 2.2. Amino Acid–Amino Acid Bond Pair (AABP) for Entire Model

Let us now shift our focus from the mutated sites to all AAs involved in the RBD–ACE2 interface model. The RBD–ACE2 interface model consists of 311 AAs, among which 194 AAs are from RBD and the remaining 117 AAs are from ACE2. The TAABP figures for the AAs in RBD and ACE2 are shown in Appendix A, respectively. These figures include comparisons among all studied models: WT, OV BA.1, OV BA.2, OV BA.5, and OV XBB.1.16. The sum of TAABP for all 311 AAs for WT, BA.1, BA.2, BA.5, and XBB.1.16 are 339.04 e^−^, 339.71 e^−^, 338.26 e^−^, 337.93 e^−^, and 334.59 e^−^, respectively. However, this sum duplicates the bond order common in bonding AAs. So, the sums of all inter-amino acid bondings for WT, BA.1, BA.2, BA.5, and XBB.1.16 are 169.52 e^−^, 169.86 e^−^, 169.13 e^−^, 168.97 e^−^, and 167.29 e^−^, respectively. In both cases, the trend is the same, with BA.1 having the highest value followed by WT, BA.2, BA.5 and XBB.1.16. This denotes that throughout the interface model, inter-amino acids are strongly bonded in BA.1, followed by WT, BA.2, BA.5 and XBB.1.16. To further assist the analysis of TAABP, their standard deviations for RBD and ACE2 are shown in Appendix A, respectively.

In Appendix A, we have presented bar graphs for total AABP (TAABP) for all 194 AAs in the RBD part of the interface model for WT, OV BA.1, OV BA.2, OV BA.5, and OV XBB.1.16, respectively. The yellow color shows the contribution from the nearest neighbor (NN), and the red color shows the contribution from non-local (NL) interaction. The first and last AAs in sequences T333 and G526, respectively, only have one NN AA interaction, reducing their TAABP values. All mutated amino acids in the RBD are marked with an arrow. We can observe differences in the pattern of TAABP throughout the sequence, especially between WT and OV BA.1. It is obvious that TAABP has a higher contribution from NN AAs in comparison to NL AAs. When comparing the changes between NN and NL AABP values between the WT and OV BA.1 in the entire sequence, we observe significant differences in the NL AABP, of up to 0.4 e^−^. Hence, the NL AA interactions are significant. Comparing Appendix A, which shows RBD WT, and Appendix A, which shows RBD OV BA.1 for NL TAABP, we observe significant increases in S366, N440K, and S477N. Moreover, a noticeable decrease in NL TAABP was observed in K417N, T333, D389, E484A, Y505H, and G526. Hence, we can conclude that the mutation sites are not the only ones with significant changes. There are in fact many other unmutated AAs that went through the changes. OV BA.2, OV BA.5, and OV XBB.1.16 show some similarities and some differences at or closer to mutation sites (see Appendix A).

For convenience, we have plotted the standard deviations of RBD for the NN and NL for all five models in Appendix A, respectively. From this, we can identify the NN and NL sequence numbers with significant changes. In the case of NN AABP from Appendix A, the sequence numbers with significant changes are 360, *375*, *376*, 378, 379, 393, 394, 425, 444, *445*, *460*, 492, *493*, and 525. Here, the italicized numbers are mutated residue sequence numbers, and the remaining are unmutated residue sequence numbers. In fact, an unmutated 379 site has the highest standard deviation for NN, showing that even unmutated residues are affected. In the case of NL AABP from Appendix A, the residue sequence numbers with significant standard deviations are 333, 364, 366, 378, 386, 394, *405*, *417*, *440*, *460*, *477*, *484*, *493*, *505*, and 526. Here, the italicized numbers are mutated sites. Overall, the NN and NL AABP standard deviations for all RBDs in all five models indicate that both mutated and unmutated AAs are impacted.

In Appendix A, we have presented bar graphs for TAABP for all 117 AAs in the ACE2 fragment of the model for WT, OV BA.1, OV BA.2, OV BA.5, and OV XBB.1.16, respectively. The dashed lines represent breaks in the sequence numbers between 19–88 and 319–365. As there is no mutation in ACE2, none of the AAs are marked. It would be interesting to see if there are any significant changes in all five models. To further analyze ACE2 bonding, let us examine the standard deviations for NN and NL AABP in Appendix A, respectively. In Appendix A, we can see a break between 19–88 and 319–365 in the *x*-axis, representing the break in the sequence number. Overall, we observe variations in both NN and NL AABP, suggesting that despite the absence of mutation in ACE2, it is still affected by the RBD mutation across the five models. Noticeable variations can be observed via the standard deviation in NN AABP for D67, K68, E75, and N338 sites (see Appendix A). Significant variations in NL AABP are observed in D30, H34, N64, E329, and K341 sites (see Appendix A). Some other noticeable sites from Appendix A are E23, K31, E35, E37, Q42, T55, E57, Q60, S70, K74, Q325, G337, N338, and V339. Especially in the case of NL AABP for ACE2, these AAs could be interacting with the mutated or AAs influenced by mutation in the RBD.

### 2.3. Bonding Between RBD and ACE2

To better understand how RBD mutation influences ACE2 binding, we analyzed the RBD–ACE2 interface for WT, OV BA.1, OV BA.2, OV BA.5, and OV XBB.1.16. The changes in the interface interaction between RBD and ACE2 across WT and four OV models (BA.1, BA.2, BA.5, and XBB.1.16) are shown in Figure 4. Figure 4a shows RBD–ACE2 interface interactions across five models involving mutation sites. The *y*-axis shows the specific interaction involving AAs from RBD and ACE2, respectively (AA from RBD-AA from ACE2). The *y*-axis represents the partial NL AABP, as there could be NL interaction within both RBD and ACE2 AAs. Let us investigate one of the strongest interactions between K417-D30 in WT. The K417N mutation is observed in all OV models, and there is no N417 interaction with D30. This could be one of the reasons that caused K417N to diminish its interaction with ACE2 [50]. Let us move into another example of N440K mutation observed in all OV models. A strong interaction among K440-E329 is observed in BA.1, BA.2, and BA.5, whereas this interaction is absent in XBB.1.16. This highlights differences even among OVs.

BA.5 were known to spread faster in mid-2022 because they were better at evading immunity from BA.1 and BA.2 infections and vaccines. Their advantage became clear as immunity from earlier Omicron began to wane [51]. There are two mutations (L452R and F486V) only observed in BA.5. Among these mutations, F486V has interface interactions with ACE2. Regarding site 486, let us discuss the WT, BA.1, and BA.2, which do not have this mutation. F486-Q24 interaction is observed in BA.1 and BA.2 but not in WT. All WT, BA.1, and BA.2 consist of F486-L79 interaction. Despite two different mutations in BA.5 (F486V) and XBB.1.16 (F486P), they include interaction with L79 of similar strength to that of the F486-L79 interaction in WT, BA.1, and BA.2. One interesting fact is that the interactions of the RBD’s 486 site with M82 and Y83 are observed in WT, BA.1, BA.2, and XBB.1.16. However, V486 of BA.5 does not interact with M82 and Y83. This could be another reason for BA.5 to be more transmissible. F486 is a critical site that interacts with ACE2 and is a key target for neutralizing antibodies [51]. The F486V mutation of the BA.5 variant was particularly concerning because it enhanced immune evasion while maintaining effective cell entry [51]. This contributed to a resurgence of infections despite previous immunity from vaccines or prior infections such as BA.1 [51].

In a recent study, mutations such as S477N, Q493R, Q498R, and N501Y significantly enhance the binding of the RBD–ACE2 interface [52]. Our study shows similar results for the four sites, which are discussed below. The mutation, S477N, is present in all OV models. In these models, N477 exhibits relatively strong bonding with S19 and weaker bonding with T20 of ACE2 in all OV models. However, in BA.1 and BA.2, N477 also interacts with Q24 of ACE2, a bonding not observed in other variants. In contrast, WT S477 lacks all these interactions.

The mutation Q493R is observed only in BA.1 and BA.2. In XBB.1.16, Q493 interacts with K31 of ACE2, an interaction absent in WT and BA.5. However, in BA.1 and BA.2, R493 interacts with K31 of ACE2. Q493 interacts with H34 of ACE2 for all WT, BA.5, and XBB.1.16, while R493 also forms this interaction in BA.1 and BA.2. Additionally, Q493 interacts with E35 of ACE2, exhibiting stronger interaction in WT and a weaker one in XBB.1.16, but it does not interact with E35 in BA.5. In contrast, R493 interacts with E35 of ACE2 in both BA.1 and BA.2. In summary, site 493 exhibits extensive interaction with three amino acids, K31, H34, and E35 of ACE2.

The Q498R mutation is present in all OV models. Both Q498 in WT and R498 in all OV models interact with D38 of ACE2, with the R498-D38 interaction being slightly stronger than Q498-D38. Regardless of the mutation, site 498 interacts with Y41 in WT and all OV models. Similarly, it interacts with Q42, with this interaction being slightly stronger in BA.1, BA.2, and BA.5. The only interaction unique to this site is Q498 with K353 in WT.

The N501Y mutation is present in all OV models. However, Y501 interacts with D38 of ACE2 only in BA.5 and with N330 of ACE2 in XBB.1.16. In both WT and all OV models, site 501 interacts with K353 and D355 of ACE2. Additionally, after the mutation, site 501 gains a new interaction with Y41 of ACE2 in all OV models.

Now, let us discuss the Y505H mutation. The Y505H mutation is present in all OV models. In WT, Y505 interacts strongly with E37 of ACE2, but this interaction becomes slightly weaker after the mutation to H505 in all OV models. Additionally, site 505 interacts with K353 of ACE2 in WT and all OV models. However, its interaction with G354 of ACE2 is observed only in WT, BA.5, and XBB.1.16, while it is absent in BA.1 and BA.2. This may be a factor that weakens RBD–ACE2 interaction [52].

Next, we highlight the interaction between RBD and ACE2. The total bond strength, calculated by summing all bond orders between interacting residues of RBD and ACE2 (including both mutated and unmutated sites), for WT, OV BA.1, OV BA.2, OV BA.5, and OV XBB.1.16, are 1.329 e^−^, 1.456 e^−^, 1.439 e^−^, 1.308 e^−^, and 1.166 e^−^, respectively. This indicates that OV BA.1 exhibits the strongest binding between RBD and ACE2 among the five models. The strongest binding between RBD and ACE2 of OV BA.1 is followed by OV BA.2, WT, OV BA.5, and OV XBB.1.16.

We would also like to point out that this study focuses solely on the bonding aspect and does not account for long-range electrostatic or van der Waals forces. The association of RBD to ACE2 is known to be pH-dependent [53], due to protonation or deprotonation of surface-exposed, chargeable amino acid residues. However, in our study, we use atomic coordinates from X-ray diffraction data of the RBD–ACE2 complex, where the protonation states of amino acids are fixed based on their pKa values and the pH conditions under which the crystal was formed. As a result, pH-dependent electrostatic effects at the RBD–ACE2 interface are beyond the scope of this work. Other studies have directly investigated electrostatic interactions and reported mutation-induced electrostatic alterations [54,55]. Nevertheless, our previous DFT-based studies have identified a mutation that alters partial charge and surface charge distribution, thereby affecting RBD’s binding to ACE2 for OV BA.1 [56,57]. We have also calculated partial charge distribution for all five models, which are presented in the section below.

### 2.4. Partial Charge for the Mutation Sites

Partial charge (PC) is a critical parameter that reflects the electrostatic potential surrounding a molecule and plays a crucial role in predicting long-range interactions [58]. Using the OLCAO method, we calculated the PC of every atom in all five models—WT, BA.1, BA.2, BA.5, and XBB.1.16—and then summed these atomic PCs to determine the PC of each residue.

For clarity, the PC results for all 311 residues at the RBD-ACE2 interface are presented separately. RBD results are listed in Appendix A for WT, BA.1, BA.2, BA.5, and XBB.1.16, respectively, while the ACE2 results are provided in Appendix A for the same variants.

Our analysis focuses particularly on the mutation sites, with the corresponding PC changes presented in Figure 5. We have identified seven mutation sites—373, 405, 440, 452, 478, 484, and 501—where PC flips from negative in the WT to positive PC in the OV. Such positively charged mutations enhance long-range electrostatic interactions with the negatively charged surface of the host cell, thereby potentially increasing binding affinity [59].

In addition, we have identified 15 mutation sites—371, 373, 405, 440, 445, 452, 460, 477, 478, 484, 486, 493, 498, 501, and 505—where PC moves toward a more positive value, either through an increase in positive charge or a reduction of negative charge relative to the WT. Compared with WT, BA.1’s 15 RBD mutations include 11 that show this positive shift; BA.2’s 16 mutations include 12; BA.5’s 17 include 11; and XBB.1.16’s 22 include 12.

While previous studies have emphasized the role of positively charged residues in enhancing electrostatic interactions [59,60], our work uniquely offers a quantitative assessment of how mutation impact the partial charge (PC) based on ab initio calculations.

## 3. Models

In this work, we have studied the RBD–ACE2 interface for five models. They are four Omicron variant models, OV BA.1, OV BA.2, OV BA.5, and OV XBB.1.16, with their corresponding Wild Type (WT) model. The WT RBD–ACE2 interface model was generated using PDB ID 6M0J [9]. It includes all RBD amino acid (AA) residues ranging from 333 to 526 AAs, and the key interacting ACE2 AA residues ranging from 19–88 and 319–365 AAs [61,62]. In this model, there are 311 AAs (4817 atoms) in total, of which 194 AAs are in RBD and 117 AAs are in ACE2.

For the OV BA.1 RBD–ACE2 interface model, PDB ID 7WBP [63] was used. The OV BA.1 RBD contains fifteen mutations, as listed in Table 1. First, we selected the 311 residues of RBD and ACE2 that matched the WT model. Then, hydrogen atoms were added using the Leap module with a ff14SB force field in the AMBER package [64]. The BA.1 model consists of a total of 4873 atoms. Selected results focusing only on the mutated AAs were published in our previous work [56,57].

The OV BA.2 RBD–ACE2 interface model was constructed based on the BA.1 model of PDB ID 7WBP and further refined according to BA.2 mutations. The RBD of BA.2 does not have mutations on G446 and G496. So, these AAs were reverted to WT. The BA.2 model has 16 RBD mutations, among which 13 are common with BA.1. The other three mutations, T376A, D405N, and R408S, are unique, which were modeled based on the high probability value of the Dunbrack backbone-dependent rotamer library [65] used in Chimera [66]. After making all the substitutions, hydrogen atoms were also added using the same approach implemented in the BA.1 model. Overall, this model also has 311 AAs, with a total of 4851 atoms.

The OV BA.5 RBD–ACE2 interface model was built using the BA.2 model (based on BA.1 PDB ID 7WBP) as a template, with the necessary modification according to its mutations listed in Table 1. Among these 17 mutations in BA.5, 15 are shared with BA.2. Since there is no mutation in Q493, it was reverted to WT. The two modeled mutations are L452R and F486V. As L452R is common with the Delta Variant, we used the PDB ID 7TEW of Delta variant to model L452R. The F486V mutation was modeled utilizing the high probability value of the Dunbrack backbone-dependent rotamer library [65] in Chimera [66]. Again, hydrogen atoms were added as in the BA.1 and BA.2 models, providing a total of 4845 atoms in the BA.5 model. It should be mentioned that both the BA.2 and BA.5 models were prepared before their PDB structures were accessible.

The interface structure of the BA.2 RBD–ACE2 complex (PBD ID:7XB0 [67]) serves as a template for generating the Omicron XBB.1.16 RBD–ACE2 model. XBB.1.16 has 22 RBD mutations. Thirteen RBD mutations are identical to those in both BA.2 and BA.5, as shown in Table 1, and thus remain unchanged when preparing the XBB.1.16 RBD–ACE2 model. The remaining nine unique mutations of the RBD XBB.1.16 (G339H, R346T, L368I, V445P, G446S, N460K, T478R, F486P, and F490S) were generated using the UCSF Chimera Dunbrack rotamer library [66]. Their substitutions were chosen using the high probability value of the Dunbrack rotamer library as well as the similar conformations of BA.2 RBD from ID:7XB0. Specifically, the dihedral angles of these nine mutations were adjusted using “Adjust Torsions” tools in Chimera to have conformations that were identical to those of the ID:7XB0 of BA.2 RBD. In addition, the Q493R mutation present in BA.2 but absent in XBB.1.16 was reverted to WT Q493 using the Dunbrack rotamer library as well. Finally, hydrogen atoms were added, yielding a total of 4832 atoms in the XBB.1.16 model. With ongoing mutations, XBB.1.16 exhibits changes in previously mutated amino acids from BA.1, BA.2, and BA.5. Three of these mutations—G339H, T478R, and F486P—are highlighted in Table 1.

## 4. Methods

Two packages based on density functional theory were employed in this study: the Vienna Ab initio Simulation Package (VASP) [68] and the Orthogonalized Linear Combination of the Atomic Orbital (OLCAO) method [69]. The combination of VASP and OLCAO has been successful in biomolecule calculations [56,57,62,70,71,72,73]. After preparing the models, we created a large cell with lattice parameters of 61 Å, 80 Å, 88 Å to ensure the interface complex was around 15 Å apart, thus avoiding the effect of periodic boundary conditions. In VASP, we implemented the projector-augmented wave (PAW) method with the Perdew–Burke–Ernzerhof (PBE) exchange correlation functional [74] within the generalized gradient approximation (GGA). An energy cutoff of 500 eV, electronic convergence of 10^−4^ eV, force convergence of −10^−2^ eV/Å, and single K-point were employed. Once the models were geometrically optimized using VASP, they were subsequently used as input in OLCAO.

The in-house-developed OLCAO package is used to calculate the interatomic interactions of the RBD–ACE2 interface. In the OLCAO, atomic orbitals are used for the basis expansion. We used a minimal basis (MB) that includes the core orbital, and orbital in valence shell of the atoms, whether occupied or unoccupied, to calculate the bonding. OLCAO determines the effective charge ${(Q}^{*})$ and bond order (BO) values based on Mulliken’s population analysis scheme [75,76]. $Q^{*}$ is the number of electronic charges associated with the atom, defined as:(1)$$Q_{\alpha }^{*}=\sum_{i}\sum_{n.occ}\sum_{i,j}C_{i\alpha }^{*n}C_{j\beta }^{n}S_{i\alpha ,j\beta }$$
where $Q_{\alpha }^{*}$ denotes the effective charge on atom α. The partial charge, $\Delta Q_{\alpha }$, is defined as the deviation of the effective charge from the neutral charge(2)$$\Delta Q_{\alpha }=Q_{\alpha }^{0}-Q_{\alpha }^{*}$$
where $Q_{\alpha }^{0}$ is the charge on the neutral atom α.

BO is the overlap population $\rho_{\alpha \beta }$ which specifies the relative strength of the bond between any pair of atoms (α, β).(3)$$\rho_{\alpha \beta }=\sum_{n.occ}\sum_{i,j}C_{i\alpha }^{*n}C_{j\beta }^{n}S_{i\alpha ,j\beta }$$
where $S_{i\alpha ,j\beta \;}$ are the overlap integrals between the $i^{th}$ orbital in the $\alpha^{th}$ atom and $j^{th}\;$ orbital in the $\beta^{th}$ atom, and $C_{j\beta }^{n}$ are the eigenvector coefficients of the $n^{th}$ band, $j^{th}$ orbital in the $\beta^{th}$ atom.

The overall BO values for all atomic pairs are used to identify the internal cohesion of the system under study [77,78]. In complex biomolecules, we have extended the concept of BO from the pair of atoms to the pair of amino acids, termed the amino acid–amino acid bond pair (*AABP*) [72].(4)$$AABP(u,v)=\sum_{\alpha \epsilon u}\sum_{\beta \epsilon v}\rho_{\alpha i,\beta j}$$
where the summations are over atoms α in amino acid u and atoms β in amino acid v. This novel concept, introduced by us in 2020 [72], accounts for all bonding between two amino acids, including both covalent and hydrogen bonding. This parameter reflects the bonding between a pair of amino acids and can be further resolved into bonding with nearest neighbor (NN) AA and non-local (NL) AA. We have further extended the concept of AABP to an AABP unit (AABPU), which includes all amino acids that interact with a selected amino acid or site. This extension provides a visual representation of how mutations alter bonding interactions.

In the present study, the AABP values were calculated for all 311 AA pairs in all five RBD–ACE2 interface models: WT, OV BA.1, OV BA.2, OV BA.5, and OV XBB.1.16.

## 5. Conclusions

In summary, we have used DFT to study the RBD–ACE2 interface of SARS-CoV-2 for five models—WT, OV BA.1, OV BA.2, OV BA.5, and OV XBB.1.16. Using DFT, we have developed the concept of AABP. From AABP, we can quantify the bonding strength between amino acids, identifying the changes caused by mutation. Using this parameter, we have analyzed all 311 AAs in the RBD–ACE2 interface across the five models, assessing the influence of RBD Omicron mutations on the interaction with ACE2 and on all unmutated sites within the RBD–ACE2 interface. In addition, we have computed partial charge for 311 residues in the five models and pinpointed those that switch from negative in WT to positive in the OVs. These positively charged mutations bolster long-range electrostatic interactions with the negatively charged host cell, potentially enhancing binding affinity. Our study delivers a quantitative evaluation of how mutation affects PC.

Certain mutated sites in RBD, such as 375, 376, 417, 440, 445, 460, 477, 484, 493, and 505, induce significant changes in bonding. Additionally, unmutated sites in RBD, including S366, T333, D364, N360, K378, C379, K386, T393, D389, N394, L425, K444, L492, C525, and G526, also exhibit noticeable alterations. Interestingly, the ACE2, despite having no mutations, is still affected by RBD mutations. The ACE2 residues showing significant variations include E23, D30, K31, H34, E35, E37, Q42, T55, E57, Q60, N64, D67, K68, S70, K74, E75, Q325, E329, G337, N338, V339, and K341. Our detailed findings provide evidence that unmutated sites are also impacted by mutation.

We have further analyzed changes in the AA bonding between RBD and ACE2 across five models. Adding up all bonding involved between RBD and ACE2, the binding is strongest in OV BA.1, followed by OV BA.2, WT, OV BA.5, and OV XBB.1.16.

We emphasize that the AABP study of amino acid interaction goes beyond simple interatomic proteins. AABP not only accurately detects changes caused by mutations but also provides deeper insights into understanding underlying mechanisms. This approach is highly applicable to large-scale computations, playing a crucial role in understanding mutation process and advancing protein design, ultimately contributing to the development of effective vaccines and therapeutic drugs [45].

We would like to highlight that while ab initio calculations are well established in materials science, their application in biophysics remains relatively uncommon. However, the detailed insights obtained from such approaches can effectively complement molecular dynamics by enhancing their quantitative predictions by providing atom-level information on bonding and identifying key interacting AAs.

## Figures and Tables

**Figure 1 molecules-30-02061-f001:**
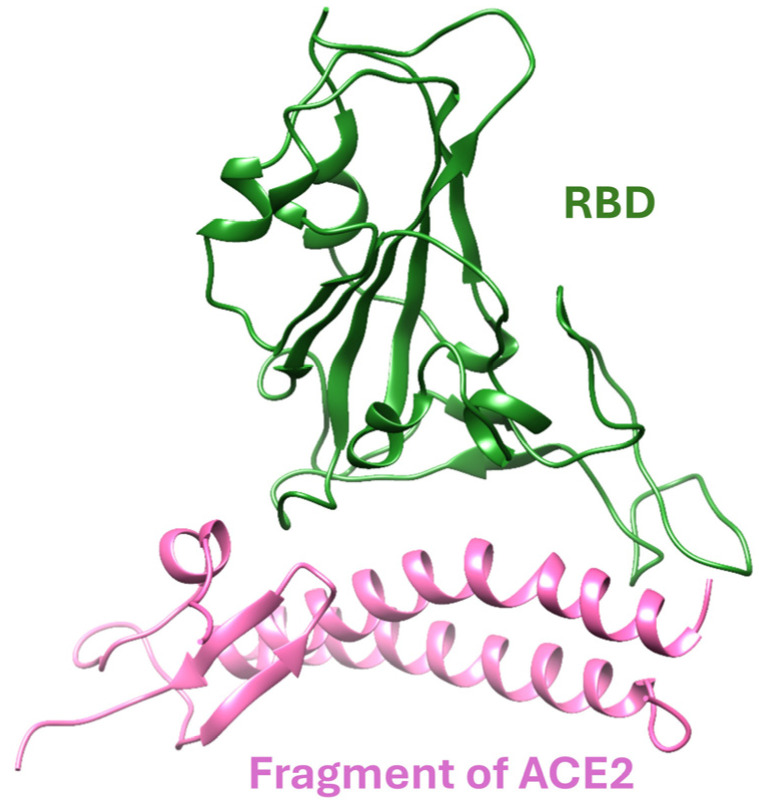
The ribbon structure of Wild Type (WT) RBD–ACE2 interface.

**Figure 2 molecules-30-02061-f002:**
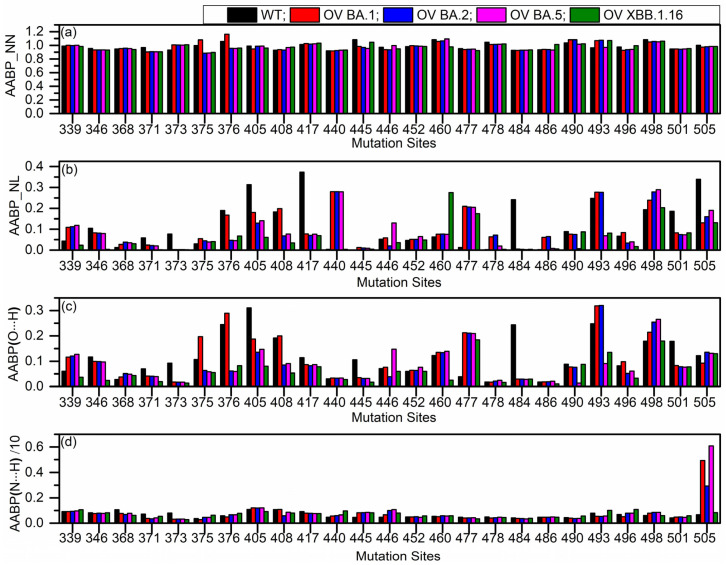
Comparison of amino acid–amino acid bond pair (AABP) analysis among Wild Type (WT) and four Omicron Variant (OV) models for (**a**) nearest neighbor (NN), (**b**) non-local (NL), and (**c**,**d**) hydrogen bonding (O⋯H and N⋯H) of mutated sites in the RBD interface.

**Figure 3 molecules-30-02061-f003:**
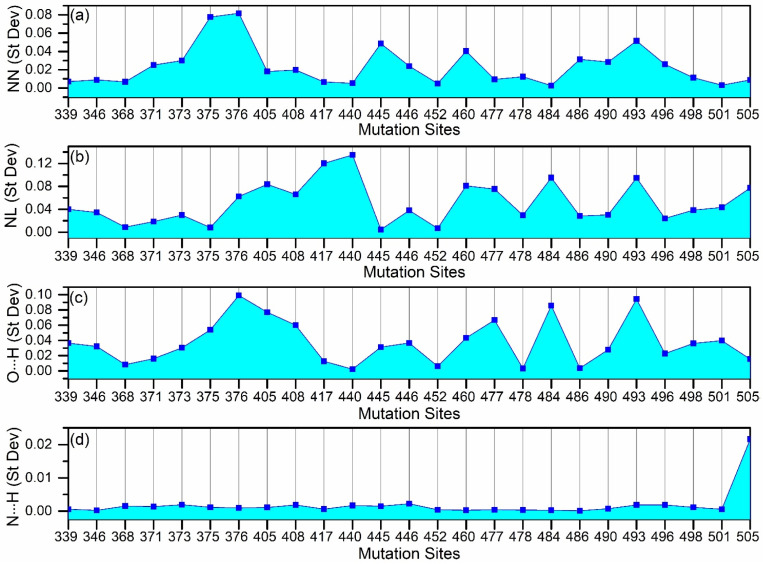
Standard deviation for amino acid–amino acid bond pair (AABP) including Wild Type (WT) and all four Omicron Variant (OV) models for (**a**) nearest neighbor (NN), (**b**) non-local (NL), (**c**,**d**) hydrogen bonding (O⋯H and N⋯H) of mutated sites of the RBD in the interface.

**Figure 4 molecules-30-02061-f004:**
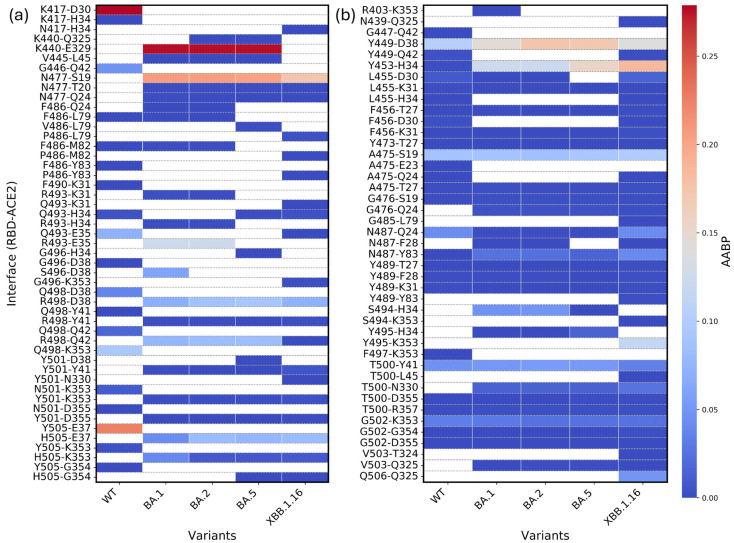
Heatmap of RBD–ACE2 interface interaction across five models—WT, OV BA.1, OV BA.2, OV BA.5, and OV XBB.1.16. (**a**) Interaction across mutation sites of RBD. (**b**) Interaction at unmutated sites of RBD.

**Figure 5 molecules-30-02061-f005:**
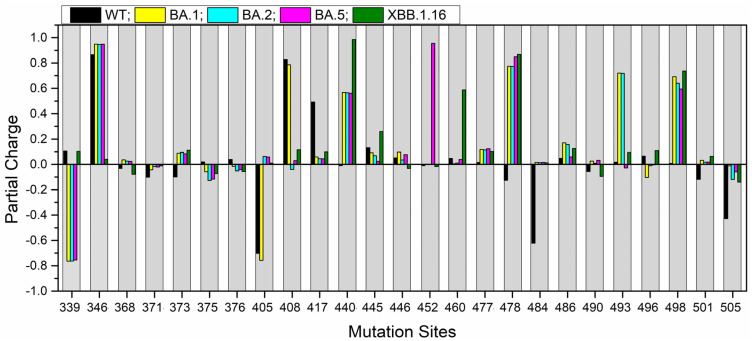
Partial charge (PC) for 25 RBD mutation sites compared across five models—WT, OV BA.1, OV BA.2, OV BA.5, and OV XBB.1.16.

**Table 1 molecules-30-02061-t001:** Mutation in OV BA.1, BA.2, BA.5. XBB.1.16.

OV BA.1	OV BA.2	OV BA.5	OV XBB.1.16
G339D	G339D	G339D	G339H
-	-	-	R346T
-	-	-	L368I
S371L	S371F	S371F	S371F
S373P	S373P	S373P	S373P
S375F	S375F	S375F	S375F
-	T376A	T376A	T376A
-	D405N	D405N	D405N
-	R408S	R408S	R408S
K417N	K417N	K417N	K417N
N440K	N440K	N440K	N440K
-	-	-	V445P
G446S	-	-	G446S
-	-	L452R	-
	-	-	N460K
S477N	S477N	S477N	S477N
T478K	T478K	T478K	T478R
E484A	E484A	E484A	E484A
		F486V	F486P
-	-	-	F490S
Q493R	Q493R	-	-
G496S		-	-
Q498R	Q498R	Q498R	Q498R
N501Y	N501Y	N501Y	N501Y
Y505H	Y505H	Y505H	Y505H

## Data Availability

Data are contained within the article or Appendix A. Further datasets are available on request from the authors.

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
