# Peer review of "Electronic Interactions Between the Receptor-Binding Domain of Omicron Variants and Angiotensin-Converting Enzyme 2: A Novel Amino Acid–Amino Acid Bond Pair Concept"

_molecules, 2025, doi:10.3390/molecules30092061_

Round 1
Reviewer 1 Report
Comments and Suggestions for Authors
This is an interesting study in which the authors use a method to quantify amino acids interactions via the amino acids-amino acids bond pair (AABP). They applied this method to investigate the interface interactions of the RBD-ACE2 complex in four Omicron Variants (BA.1, BA.2, 21 BA. 5, and XBB.1.16) with its Wild-Type counterpart.
I will only comment on the methodology used by the authors. In this sense, I think that the authors should present the methodology section in a more in-depth way, answering questions like the following
- How do you ensure a supercell size of 15 is sufficient to avoid periodic boundary conditions?
- Are the numerical parameters used in the calculations sufficient to ensure adequate electron density convergence?
- Were the structures geometrically optimized?
Finally, it would be valuable to include figures showing the shape of AABPU of some selected amino acids-amino acids pair.
Author Response
The response in attached in the word file.

Reviewer 2 Report
Comments and Suggestions for Authors
The manuscript "Interface interaction of RBD Omicron variants with ACE2 receptor based on Amino acids-Amino acids bond pair as a novel concept in biomolecule interaction" in an interesting attempt to quantify the interaction between one of the crucial proteins of SARS-CoV-2 and ACE2 receptor. The OLCAO approach seems at first like an interesting idea and method of obtaining more information about the interface in question. Unfortunately, upon reading the entire manuscript I must say that there are many flaws, which make this study unsuitable for Molecules journal. First, authors are relying on static crystal structures and some of the models are prepared on the basis of computations of rotamers (for selected mutations), which are not validated. I'm pretty sure that even a short MD of the system will quite drastically change some of the interfaces and, in fact, to get a better picture, such analysis should be repeated for multiple MD snapshots to obtain a more accurate view of interactions.
The bigger problem is that the results are somewhat trivial. The crucial mutations can be identified within seconds just by looking at the crystal structures of mutants and this detailed analysis of AABP leads to identical conclusions. Using modern methods one could actually quite fast get the actual interaction energies between the most crucial amino acids and even better quantify the interactions at interfaces. In this regard, the proposed computational approach does not really provide anything except for some obvious results. Here, the flaw is in the design of the entire study and the method used.
Finally, the description of the results reads quite bad. See e.g. this part: "NN AA is the most important as it follows the primary structure of the protein. However, even with the overall lower AABP contribution, NL AAs are equally important. NL interactions show the twists and turns in the protein structure. NL AABP provides insights beyond what only mere protein sequence can reveal." which does not make much sense.
Minor points:
The introduction is too basic and instead of describing the general structure of proteins should focus on the SARS-CoV-2 and ACE2 receptor.
The RBD abbreviation should not be used in the title, as it is not that common.
Author Response
The response is attached.

Reviewer 3 Report
Comments and Suggestions for Authors
The authors tried to investigate the interactions of the receptor biding domain (RBD) of the S-protein of the Omicron variant of SARS-CoV-2 coronavirus and its five subvariants with the epithelial cell receptor angiotensin converting enzyme-2 (ACE2) with the aim to estimate the association strength of the two proteins in the RBD-ACE2 complex. For the purpose, they computed the bonding strength between all i- and j-atoms at the RBD-ACE2 interface by the density function theory of the atomic orbitals developed in quantum mechanics. The atom pairs are distinguished into two types: “neighbour” and “non-local”, 311 amino acid pairs are included in the calculations. The atoms are grouped into amino acids, an approach developed by authors to calculate pairwise interaction between the opposite amino acid residues in the RBD-ACE2 interface. The two protein surfaces in the RBD-ACE2 complex are on 1.5 nm distance; this allows computing the O…O and N…O hydrogen bond in units of electron. The results in the supplementary file show that the first type hydrogen bonds are in order of magnitude stronger than the second one at the most mutations in RBD. Ten mutation sides and the fifteen unmutated sides are then demarcated on RBD surface of the five Omicron subvariants, and compared to the wild type Omicron RBD. Besides the quantified bonding strength, an interesting result is that the unmutated sides and the corresponding sites of ACE2 receptor are affected by the mutations in the viral RBD.
The main imperfections of this investigation are two. Firstly, the applied research approach only allows computing the hydrogen bonds in the protein-protein interface, but not other type of forces such as electrostatic and van-der-Waals ones. The S-protein and ACE2 are oppositely charged and therefore the electrostatic attraction gives the main contribution in the RBD-ACE2 complex. Even between identical protein globules, an association can form due to the irregular distribution of the charged amino acid residues on the globule surface leading to formation of regions with positive and negative electrostatic potentials. A formation of electrostatic dimers and trimers by identical cytochrome globules is shown in Int. J. Mol. Sci. 25 (2024) 6834; in the same article it is revealed that the van-der-Walls attraction plays a secondary role. Secondly, the authors’ computing disregards the fact that interactions between RBD and ACE2 are pH-dependent because of the different degree of proton association or dissociation of the chargeable groups of the superficially located seven types amino acid residues. In the literature the electrostatic interaction between the RBD of the S-protein and ACE2 was investigated by two approaches: (a) by pairwise summation of the electrostatic energy of the charged opposite groups in the RBD-ACE2 interface (J. Phys. Chem. B, 126 (2022) 6835-6852); and (b) calculation of the association energy between the oppositely charged surfaces of RBD and ACE2 (Viruses, 15 (2023) 1752). The electrostatic alterations caused by point mutations in Omicron RBD are investigated in: J. Biomol. Struct. Dynamics, 41 (2023) 5707-5727 and in: Int. J. Mol. Sci. 25 (2024) 2174.
Recommendations:
- Change the title of the manuscript because the present one does not adequately reflect the content of the investigation. A possible variant is “Hydrogen bonds in the interface of ACE2 receptor and RBD of S-protein of Omicron variant of SARS-CoV-2 coronavirus and five its subvariants”.
- Describe how RBD and ACE2 are mutually oriented. Is an experimental model (crystallographic, NMR, cryoEM) used, or the two protein globules are oriented by docking?
- Remove the first paragraph in the Conclusion; this text is suitable for the Introduction.
- Cite the above mentioned articles of Barroso da Silva and S. Hristova.
- The text in Results describing the results including in the “Supplementary materials” should be removed to the last file.
- It seems, the standard deviation (Figure 3) and the corresponding text are on secondary importance and could be removed to “Supplementary materials”.
Author Response
The response is attached.

Reviewer 4 Report
Comments and Suggestions for Authors
The paper is well-written and presents a compelling study on RBD-ACE2 interactions. I have just some minor recommendations to improve the clarity of the manuscript:
lines 52-63: excessive citation of one paper. Please, add another references.
line 67: Please, clarify which structural data you mean (primary, secondary, tertiary, quaternary structure of protein or other data).
line 80: add reference (https://doi.org/10.1016/j.cell.2020.02.052 or other).
line 91: add reference to the WHO site.
Please provide additional details on standard deviation calculation.
Clarify which AABP (NN, NL... or total) you mean in Fig.4.
Author Response
The response is attached.

Round 2
Reviewer 3 Report
Comments and Suggestions for Authors
The revised manuscript needs additional improvement as follows.
- The revised title "Interface bonding ..." remains deceptive because the driving forces of protein-protein attraction are long-range. They are caused predominantly by the surface electrostatic potential that originated from the coulomb charges of the superficially located amino acid residues. The contribution of the partial charges is insignificant because their origin is from polarized covalent bonds which can be assumed as dipoles whose electric field rapidly decreases with the distance. The field of the partial charges is essential only on a distance commensurable with the length of the covalent bonds. In this research the short-range acting electron interactions are investigated, which lead to formation of hydrogen bonds. They do not give an essential contribution to the association energy of RBD-ACE2 complex because the attraction between the two protein globules is determined predominantly by the electrostatic forces originating from the coulomb charges. That is why the title should not be "Interface bonding ..." because the authors do not calculate the long-range electrostatic and London dispersion forces. I think that the main result in this research is the calculation of the O…O and N…O hydrogen bonds in units of electron. That is why the "Interface bonding ..." in the title should be replaced with “Hydrogen bonding” or with “Electronic interactions …”.
- The addition on lines 385-393 is useful but insufficient because the coulomb charges, which give the main contribution to the binding energy, are pH-dependent. To remove this imperfection an additional text should be added like this: “… the association of RBD to ACE2 is pH dependent [Viruses, 15 (2023) 1752] because of protonation or deprotonation of the chargeable groups of the superficially located amino acid residues. However, we compute the electronic interactions between the amino acid residues of the two protein macromolecules whose atomic coordinates are determined by X-ray diffraction of RBD-ACE2 crystal where the chargeable groups have fixed charge; i.e. they are protonated or deprotonated according to their pKa values and pH of the medium in which the protein crystal is grown. That is why the pH-dependent electrostatic interactions on RBD-ACE2 interface are beyond the scope of this work.”
- The new text in Introduction (the beginning of page 3, lines 83-94) is partially incorrect because the signal sequence is not included in the maturated protein. Therefore, the word ”immature” should be added: “The immature S1 subunit includes…”. The role in ACE2 is not fully discusses, information that it also breaks down Ang I into Ang (1-9) is missing.
Author Response
The response letter is attached.
